# Evaluation of Training with Elastic Bands on Strength and Fatigue Indicators in Paralympic Powerlifting

**DOI:** 10.3390/sports9100142

**Published:** 2021-10-12

**Authors:** Felipe J. Aidar, Filipe Manuel Clemente, Luiz Fernandes de Lima, Dihogo Gama de Matos, Alexandre Reis Pires Ferreira, Anderson Carlos Marçal, Osvaldo Costa Moreira, Alexandre Bulhões-Correia, Paulo Francisco de Almeida-Neto, Alfonso López Díaz-de-Durana, Eduardo Borba Neves, Breno Guilherme Araújo Tinoco Cabral, Victor Machado Reis, Nuno Domingos Garrido, Pantelis Theo Nikolaidis, Beat Knechtle

**Affiliations:** 1Group of Studies and Research of Performance, Sport, Health and Paralympic Sports (GPEPS), Federal University of Sergipe (UFS), São Cristovão 49100-000, Brazil; fernandes_aju93@hotmail.com (L.F.d.L.); dihogogmc@hotmail.com (D.G.d.M.); alexandreispf@gmail.com (A.R.P.F.); acmarcal@yahoo.com.br (A.C.M.); 2Department of Physical Education, Federal University of Sergipe (UFS), São Cristovão 49100-000, Brazil; 3Graduate Program of Physical Education, Federal University of Sergipe (UFS), São Cristovão 49100-000, Brazil; 4Escola Superior Desporto e Lazer, Instituto Politécnico de Viana do Castelo, Rua Escola Industrial e Comercial de Nun’Álvares, 4900-347 Viana do Castelo, Portugal; flilipe.clemente5@gmail.com; 5Instituto de Telecomunicações, Delegação da Covilhã, 1049-001 Lisboa, Portugal; 6Cardiovascular & Physiology of Exercise Research Laboratory, Faculty of Kinesiology and Recreation Management, University of Manitoba, Winnipeg, MB R3T 2N2, Canada; 7College of Physical Education and Exercise Science, University of Brasília (UnB), Brasília 70910-900, Brazil; 8Institute of Biological Sciences and Health, Campus Florestal, Federal University of Viçosa, Viçosa 35690-000, Brazil; ocostamoreira@gmail.com; 9Department of Physical Education, Federal University of Rio Grande do Norte (UFRN), Natal 59078-970, Brazil; alexandrebulhoescorreia@gmail.com (A.B.-C.); paulo220911@hotmail.com (P.F.d.A.-N.); brenotcabral@gmail.com (B.G.A.T.C.); 10Sports Department, Physical Activity and Sports Faculty-INEF, Universidad Politécnica de Madrid, 28040 Madrid, Spain; alfonso.lopez@upm.es; 11Graduate Program in Biomedical Engineering, Federal Technological University of Paraná (UTFPR), Curitiba 80230-901, Brazil; eduardoneves@utfpr.edu.br; 12Research Center in Sports Sciences, Health Sciences and Human Development (CIDESD), Trásos Montes and Alto Douro University, 5001-801 Vila Real, Portugal; victormachadoreis@gmail.com (V.M.R.); ngarrido@utad.pt (N.D.G.); 13School of Health and Caring Sciences, University of West Attica, 12243 Egaleo, Greece; pademil@hotmail.com; 14Exercise Physiology Laboratory, 12243 Nikaia, Greece; 15Institute of Primary Care, University of Zurich, 8091 Zurich, Switzerland; beat.knechtle@hispeed.ch; 16Medbase St. Gallen Am Vadianplatz, 9001 St. Gallen, Switzerland

**Keywords:** Paralympic powerlifting, elastic bands, variable resistance, muscle strength, physical education and training

## Abstract

Background: Variable resistance training has recently become a component of strength and conditioning programs. Objective: This randomized counterbalanced cross-over study aimed to investigate the use of elastic bands (EB) and the traditional method (TRAD) and force indicators in a training session. Methods: 12 Paralympic athletes (age: 28.60 ± 7.60 years) participated in this three-week study. In the first week, the participants were familiarized with EB and TRAD and were tested for maximal repetition (1-RM). The research occurred in weeks 2 and 3, which included the pre-post training, during which the following measures were extracted: maximum isometric force (MIF), the peak torque (PT), rate of force development (RFD), fatigue index (FI), and time to MIF (Time). The athletes performed two tests, EB and TRAD, separated by a one-week interval. Results: Significant differences were found between the pre- and post-test for 1RM (*p* = 0.018, η2p = 0.412), MIF (*p* = 0.011, η2p = 0.415), PT (*p* = 0.012, η2p = 0.413), and RFD (*p* = 0.0002, η2p = 0.761). With the use of EB, there was a difference in RFD between TRAD before and EB after (*p* = 0.016, η2p = 0.761). There were significant differences in the before and after for FI between TRAD and EB (*p* < 0.001) and for Time (*p* < 0.001), indicating that training with the use of elastic bands promotes overload, characterized by increased fatigue and decreased strength. Conclusions: Training with EB did not decrease 1RM, PT, MIF or RFD, however, there was an increase in fatigue and time to reach MIF when compared to the method with fixed resistance.

## 1. Introduction

Strength training is used for the physical conditioning of athletes and recreational individuals, as it can generate improvements in strength, hypertrophy, motor performance, and body composition [1,2,3,4]. In this context, variable resistance training is an exercise method that allows changes in applied resistance across the entire range of motion, used to increase strength [5,6], especially during traditional training with fixed resistance. The force exerted by the muscle varies according to the joint range of motion as the length of the arm of the muscle force varies according to the joint’s angular position.

In this context, we emphasize that the powerlifting modality promotes an improvement in performance for Paralympic athletes, as well as gaining more and more followers. [7]. This Paralympic sport was adapted from powerlifting for people with physical disabilities (especially in the lower limbs) [8]. Several studies have investigated training methods that use variable resistance, mainly with elastic bands, for sports that depend on strength [5,9] or to increase strength in the horizontal bench press [10,11]. Training with elastic bands can promote body instability when performing exercises such as bench presses, causing an inverse resistance towards the ground [12,13]. This instability may require greater recruitment of motor units to perform the desired movement [12,13]. Thus, elastic bands can promote muscle strength adaptations [10,14].

Elastic bands effectively promoted increased overload during the execution of push-ups, thus promoting muscle strength gains similar to those obtained by the horizontal bench press [15]. Similar results were observed in the muscle activation of the deltoid when comparing exercises with dumbbells to those performed with elastic bands [16]. In addition, previous studies have evaluated the effects of training with elastic bands in interventions with intervals of eight to 12 weeks [16,17,18,19], and few studies have addressed the acute evaluation of the use of this device [20,21]. Joy et al. [14] showed that when using resistance in a periodized training program, power and strength were increased in basketball athletes. Nevertheless, the use of elastic bands in resistance training promotes neuromuscular adaptations for performance and tends to increase fatigue after training [16,22,23,24].

Therefore, the present study hypothesizes that among Paralympic athletes, the fatigue index—a loss of strength during specific tests—tends to be higher after exercising with variable resistance when compared to traditional training. The characterization of fatigue and performance impacts promoted by elastic bands during training sessions could provide useful information for coaches. Specifically, it could help coaches in determining the most effective periods of exertion and the most appropriate recovery strategies to ensure optimal performance while lifting. Despite the importance of understanding such an impact, as far as we know, no study has been conducted on Paralympic athletes, which is needed for a proper understanding of the use of elastic bands among this population. Consequently, this study aimed to assess the effect of a training session using elastic bands vs. a training session with fixed resistance, through evaluation in static and dynamic indicators of force and fatigue in Paralympic athletes.

## 2. Materials and Methods

### 2.1. Participants

The sample consisted of 12 Paralympic powerlifting (PP) athletes, and as inclusion criteria, they should have had at least 12 months of training. All participants were national-level competitors and met the criteria established by the International Paralympic Committee (IPC) [8]. All athletes were ranked among the top 10 places across the country, with two national champion athletes, one runner-up, three placed third, and also are considered elite athletes due to their maximum relative strength being above 1.4 times the body mass [25]. Among the deficiencies, five athletes had spinal cord injury below the eighth thoracic vertebra, four had motor impairment due to polio, and three had poorly formed lower limbs. Exclusion criteria are defined as features of the potential study participants who meet the inclusion criteria but present with additional characteristics that could interfere with the success of the study or increase their risk for an unfavorable outcome. The athletes belong to the project called Powerlifting Paralympic, developed for the Department of Physical Education at the Federal University of Sergipe (UFS), and the Group of Studies and Research on Performance, Sport, Health and Paralympic Sports (GPEPS), from the same university. It should be noted that the UFS team is the fourth best in Brazil, having athletes with medals in international competitions and participation in games for Pan-Americans and Paralympics. The athletes volunteered to participate in the study and signed the free and informed consent term, according to resolution 466/2012 of the National Research Ethics Commission (CONEP) of the National Health Council and following the ethical principles Declaration of Helsinki (1964, reformulated in 2013), by the World Medical Association. This clinical trial was previously registered in the database of the Research Ethics Committee of the Federal University of Sergipe (ID-CAAE: 71549517.0.0000.5546), obtaining approval and ethical registration from the local ethics committee (technical advice: 2,637,882). The subjects participated in three to five training sessions per week, each one lasting three hours, between bench press and support exercises (Table 1).

The sample size was calculated, a priori, using the “F” statistic, considering the variable rate of force development (RFD) acquired by the 1RM test performed on the bench press by the 12 participants. The obtained effect size was η2p = 0.750; thus, we considered an α < 0.05 and a standard β of 0.80, with a sample power of 0.80 (strong) being estimated for the minimum number of 12 individuals for the present study. Subsequently, after the training sessions, the sampling power was calculated a posteriori, and we identified a η2p = 0.761 for the RFD variable, indicating a sampling power of 0.84 (strong) for the sample of the present study, a result that was convergent with the calculation a priori carried out before the training sessions. The open-source G* Power^®^ software (Version 3.0; Berlin, Germany) was used to calculate the sampling power.

### 2.2. Experimental Design

This study followed a randomized counterbalanced cross-over design. Through static and dynamic strength tests, we analyzed the effects of two different training methods (i.e., traditional and with elastic bands) on the performance of PP athletes. The study lasted three weeks. All tests were performed on different days at the same time (between 9:00 a.m. and noon) at temperatures ranging between 23 and 25 °C with a relative humidity of ~60%. All tests were performed on an adapted bench press [4].

In week 1, a familiarization session with the elastic band (EB) and traditional training (TRAD) methods, followed by a 1 Maximum Repetition test (1RM) was performed. During weeks 2 and 3, athletes were randomly allocated to one of two types of training: EB or TRAD. In this way, the sample was divided into two conditions with the same number of participants (50%) each week.

In the second and third weeks, the athletes were obtained by training 5 sets of 5 repetitions (5 × 5), with elastic bands or fixed resistance, and before and after the training session, the maximum Isometric Force (MIF) with peak torque measurement (PT), the rate of force development (RFD), the fatigue index (FI), and the time in the maximum isometric force (Time) were evaluated. For PT, the bar remained 15 cm above the chest, and the elbows remained at an angle of 90°. Before each experimental test, each participant remained seated for 10-min [26,27]. All tests and measures described above were also performed at the end of the training sessions period. The athletes were instructed to avoid strenuous exercise and not to drink caffeine for 48 h before the tests (Figure 1).

### 2.3. Procedures

For the analysis of body mass, a specialized digital electronic scale was used to weigh individuals in a wheelchair (Model Mic Welcair^®^; Michetti, São Paulo, Brazil. Dimension 1.02 × 1.20 m^2^; maximum capacity of 500 kg).

Rubber strips with a load of 20% of 1 Repetition Maximum (1RM) were used in a digital dynamometer (Data Weighing Systems, Wood Dale, IL, USA) with a maximum weight of 150 kg, 0.05 kg resolution, weighing modes in Lb and Kg, with dimensions of 14 × 8 × 4.5 cm^3^ (L × W × H).

For the tests, a bench press (Eleiko^®^, Halmstad, Sweden), approved by the International Paralympic Committee (IPC) was used [8]. The bar used was also following the standards of the International Paralympic Committee (weight of the bar: 20 kg, length: 220 cm with markings on narrow and wide footprints ranging from 42 cm to 81 cm) [8].

The training session consisted of five sets of five repetitions (5 × 5) [27], with full range of motion, with the differences between the training sessions being the use of elastic bands or fixed resistance bands. All athletes already had experience both in training with elastic bands and in training with fixed resistance (Figure 2).

### 2.4. Force Measurements

To measure the components of muscle force, MIF (N), PT (N.m), RFD (N·s^−1^), FI (%), and time to MIF (s), a Chrono force sensor (Chrono jump^®^, Barcelona, Spain) was used. The device has a capacity of 500 kg, an output impedance of 350 ± 3 ohms, insulation resistance >2000 cc, input impedance 365 ± 5 ohms, and digital-analog converter 24-bit and 80 Hz. The force sensor was fixed to the adapted Bench with Spider HMS Simond carabiners (Sigmond Chamonix, Chamonix Mont-Blanc, France). A steel chain with a load of rupture of 2300 kg was used to fix the force sensor to the bench. The perpendicular distance between the force sensor and the center of the joint was determined and used to calculate joint torques and fatigue index [27,28,29,30,31].

The maximum torque generated by the upper limbs was taken as the peak isometric torque (PT). We determine the PT through the product of the peak isometric force, measured between the attachment points of the force sensor cable and the adapted bench. The angle of the elbows should be close to ≅ 90°, with a 15 cm distance from the starting point (from the chest to the bar). The consistency between the angle of flexion of the elbow at the starting point over the three different trials [32] was checked with an FL6010 goniometer (Sanny^®^, São Bernardo do Campo, Brazil) (Figure 3). The force sensor was fixed 30 cm below the participant’s chest and the bar was 45 cm from the force sensor. For the acquisition of PT, the participants performed a single maximum movement with total elbow extension. The fatigue index (FI) was assessed with the same exercise; thus, the individuals maintained the maximum contraction for 60 s. The FI was calculated as FI = {(final PT-initial PT/final PT) 100}. The MIF was measured considering the maximum isometric force generated during the same movement, and the RFD was determined using the force-time relationship until reaching the maximum force (RFD = Δforce/Δtime) in 300 ms [14,21,27,28,29,30,31].

A 1RM test was performed, and this record was used to establish the load for the traditional method training session. Each individual chose their initial load, and load increments were added until the maximum lift was reached. Whenever an unsuccessful attempt occurred, the load was reduced by 2.4% to 2.5%. Participants rested for 3 to 5 min between trials. The test for determining 1RM was performed 72 h before the training sessions [27].

Three trials were made in the measures of MIF, PT, RFD, FI, and Time in MIF. In these tests, the bar was maintained at 15.0 cm above the chest, and it was necessary to apply maximum force for 5.0 s, except for the FI. For FI assessment, participants continued to perform maximum isometric contraction for one minute, and loss of PT was verified between the start of the test and up to 60 s after the beginning. The tests were performed before and after the training sessions, with a minimum interval of 10 min between tests and training. [26,27,28,29,30,31]. In warming up, the procedures were used in accordance with other studies of our group through global warming, and a gradual increase in load and specific exercises [33].

### 2.5. Elastic Bands

Participants were placed in a loop of the Eleiko brand Paralympic bench (Eleiko, Halmstad, Sweden) on the bank racks. The athletes held the bar with their elbows extended, and with the bar in contact with the chest. This distance was measured to ensure that only the resistance produced by the bands was measured. After, the athletes placed the bar on the support, and the resistance produced only by the bands in the upper and lower position was measured. The average resistance of the elastic bands over the entire range of motion was as close as possible to 20% of the workload [23]. The athletes underwent training sessions in which five sets of five RM (5 × 80–90%RM) (80–90% = 20%1RM Elastic + Weight on the bar) were performed with two elastic strips (variable resistance) attached at the end of the bar with a quantified load as close as possible to 20% of 1RM, during the range of motion [23].

### 2.6. Statistics

To check the normality of the variables, the Shapiro-Wilk test was used. The two-way ANOVA (Moments X Training) with Bonferroni’s post hoc was used to compare the performance between the training methods. The effect size was evaluated with the partial eta square (η2p), considering magnitude: low < 0.05, average 0.05 and <0.25, high 0.25 and <0.50, and very high ≥0.50 [34]. Statistical analysis was performed using the Statistical Package for Social Science (SPSS^®^), version 20. For all analyzes, the level of significance was set at *p* < 0.05.

## 3. Results

Table 2 and Figure 4 show the results found concerning training using elastic bands and the traditional method. Regarding the posterior sampling power, we highlight that, for all variables, the power was >0.80 (strong).

Table 2 shows that there were significant differences between the pre and post-test moments for 1RM, MIF, PT, and RFD. Regarding the use of EB, there was a difference in the RFD between TRAD Before and EB. For the other variables, there were no significant differences in the use of EB.

Significant differences in FI were found pre- and post-session in the traditional training method (22.05 ± 5.83%, 95%CI: 18.35–25.75 vs. 75.03 ± 7.52, 95%CI: 70.25–79.80), as well as in EB (23.57 ± 4.50, 95%CI: 20.71–26.43 vs. 85.49 ± 6.83, 95%CI: 81.14–89.43), with a very high effect (η2p = 0.983). Differences were also found between TRAD and EB post-training (*p* = 0.002, F = 13.61, η2p = 0.553), with a very high effect (see Figure 4).

Regarding the time until the MIF, significant differences were found in the fatigue index before and after the session in the traditional training method (0.71 ± 0.32 s, 95%CI: 0.56–0.85 vs. 1.10 ± 0.20 s, 95%CI: 70.25–79.80), as well as in EB (0.61 ± 0.47 s, 95%CI: 0.31–0.91 vs. 1.22 ± 0.26 s, 95%CI: 1.08–1.42), with a very high effect (η2p = 0.784). The TRAD and EB methods also showed differences in post-training (*p* = 0.033, F = 3.23, η2p = 0.227), with a medium effect.

## 4. Discussion

The present study aimed to investigate changes in strength indicators generated by training with and without EB strength training methods in a single training session among PP athletes. The study revealed that a single training session with EB and traditional resistance improved the strength indicators in the bench press among the PP athletes, without significant differences in terms of PT.

Studies involving PP are rare, so our discussion will focus on research involving similar subjects. Most studies have shown positive results with variable resistance in long-term training [10,24]. Although variable resistance tends to offer more stimuli, thus promoting greater resistance and adaptations of strength for health [35,36]. However, in the present study, during the post-workout period, EB tended to produce more fatigue than traditional training.

Bellar et al. [10] found that the combination of elastics with fixed resistance was superior to the traditional method, producing an improvement in strength increase for 1RM in the bench press. This increase in strength can be explained by increased neural action, by the use of EB [37,38].

In previous research, variable resistance associated with elastic methods did not show differences for the three different conditions [38] and was used in other studies in-volving the bench press exercise [39], corroborating with us. However, a meta-analysis [37] found that training with EB or with the use of chains combined with fixed loads, was effective in long-term training (≥7 weeks), producing positive effects on 1RM (mean differences of weight = 5.03 kg, 95% confidence interval: 2.26–7.80 kg, Z = 3.55, *p* < 0.00) and improving strength development, muscular coordination, and the recruitment of motor units, while reducing the force drop in deficient points.

There were no differences between the methods in PT. However, according to Hintermeister et al. [40], in an electromyographic evaluation, there was an increase in strength, EB in relation to fixed weights, ranging from 18 N to 54 N. Likewise, Wallace et al. [41] showed increased peak torque and mean strength with variations in resistance during a range of motion. Rivière et al. [23] compared the strength and power adaptations in response to training with variable resistance or traditional resistance training, where there were strength gains in both training methods.

Regarding fatigue, the method with EB presented a higher fatigue index than the traditional method (55.47 ± 14.61 and 30.40 ± 21.14, respectively). According to Lorenz [42], the overload is provided by the combination of variable resistance and fixed resistance. Variable resistance tends to combine amplitude with acceleration, promoting load increases over the traditional strength training method [37], making this data important for tracking the training of powerlifting athletes.

EB have been used in strength training to create extra resistance in the concentric phase. Consequently, they increase the effects on the shortening-stretching cycle [43], which tends to cause greater adaptations and fatigue in athletes. Increased variable resistance places greater demands on muscle strength generation over the range of motion [44], generating more stress than invariant resistance methods. When stretching the EB, more force is needed to overcome the resistance of this device. This procedure tends to increase delayed muscle pain [45,46].

Regarding RFD, it has been used to assess the rapid onset of strength and fatigue. RFD indicates a decrease in mechanical muscle function after injuries and strenuous training [47,48]. The RFD would be the relationship between force variation and time and is measured as the maximum value reached in a time between 100 and 300 ms [47,48]. The RFD tends to be complex, as it is influenced by several physiological and methodological aspects, involving rapid muscle contraction [49,50]. In our study, the RFD was lower in the EB group; however, as there was no decrease in maximum strength (1RM), this was probably due to increased fatigue, which decreased the athletes’ ability to generate force quickly.

As mentioned, the training session was performed as full range of motion, using variable resistance (elastic bands) or fixed resistance, and tests were performed with a force sensor before and after the training session. The indicators used, such as RFD, aim to observe the force production in the shortest possible time, as well as the MIF and Time, evaluate the force production in a given time [27,28,29]. In this sense, they are recognized as part of the force-time curve, which determines that the higher the RFD values obtained, the greater the athlete’s ability to generate force in a given time, the MIF, refers to the greater force generated, the Time the time to maximum isometric strength and fatigue would be what the athlete lost strength in a given time [30,31,51,52]. Thus, training with elastic bands was shown to promote greater fatigue, with a decrease in the main strength indicators, and this should be taken into account in the training planning by paralympic powerlifting trainers.

In relation to the RFD, a review that evaluated 47 studies showed that the strength-time characteristics in isometric tests would have a very strong correlation with the dynamic performance of the upper limbs and with the performance of sports movements, suggesting that these isometric tests would be related the strength production capacity of athletes, and in relation to dynamic performance [53]. The RFD would have high reliability in relation to the upper limbs and in multiple joints [54,55]. RFD are normally determined through isometric tests, which allow greater control of the joint angle and angular velocity changes [56].

On the other hand, RFD seems to be related to sports performance and functional daily tasks [55,57], to acute and chronic alterations in neuromuscular function [58,59,60] and physiological mechanisms [61,62].

The RFD is related to muscle fiber type mainly type II [63,64]. In the same direction, RFD would be related to neural adaptations, functional hypertrophy and increase in tendon stiffness, induced by training [65]. Training with higher loads (>75% of 1RM) appears to cause increases in RFD [47,66]. Thus, the RFD would be a very important instrument in the assessment of muscle response in terms of strength indicators, as used in our study. However, increases in RFD have been reported after intense strength training with high loads (>80% of 1RM). This gain would be explained by increases in the efferent neural impulse, due to the increase in the electromigratory signal and RFD [47], which would explain our findings.

Our results showed an increase in RFD, PT and FIM greater in the traditional training and less with the use of EB. On the other hand, fatigue was greater with the use of EB, compared to traditional training. This could be explained by the fact that for longer duration contractions (>75 ms), the RFD would be more influenced by the properties related to muscle velocity and strength in relation to the maximum voluntary contractions (MVC) [62,67]. With regard to the MVC, this would be greater at the beginning of the contraction and would still show great variability, especially in fast contractions [67,68,69]. Thus, this variability tends to reduce during a single training session [70,71], with increases in acceleration, explained by increased corticospinal excitability (43 and 63% after 150 and 300 repetitions) [72]. From what can be inferred that training would promote an increase in strength indicators, however, this tends to increase the fatigue associated with training.

A limitation of this study is that the participants obey the IPC functional classification. All competed in the same category and the possible specificities of each disability were not considered. Furthermore, overall force production was not measured during training. EB are used to increase (or at least maintain) strength throughout the range of motion. Thus, some neuromuscular adaptations may occur due to the intervention’s activation pattern, especially with variable resistance. Finally, although participants were instructed to maintain their previous regular lifestyles, particularly with regard to their diet, strict control is impossible. Therefore, similarly to other studies involving neuromuscular training, this can be considered a limitation.

## 5. Conclusions

The results show that training using elastic bands tends not to decrease the 1RM, PT, MIF, or RFD of athletes. However, it increases fatigue and enhances the time needed to reach maximum isometric strength when compared to traditional methods that involve fixed resistance. In this sense, the present findings reveal important indications that should be considered when planning and controlling the load imposed on athletes to allow for the necessary recovery according to the type of training used.

## Figures and Tables

**Figure 1 sports-09-00142-f001:**
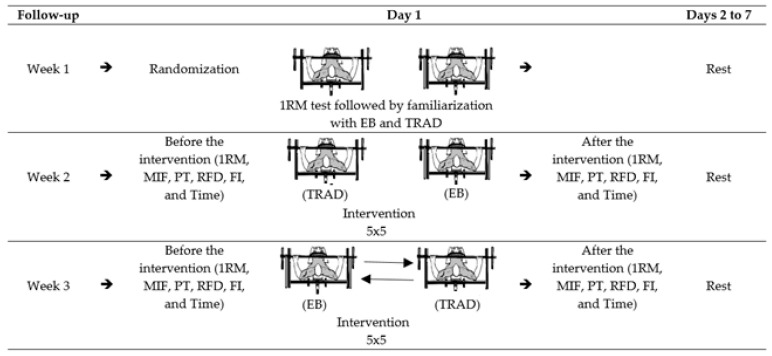
Experimental design. Legend: 1RM: 1 Repetition Maximum. MIF: Maximum Isometric Force. PT: Peak Torque. RFD: Rate of Force Development. FI: Fatigue Index. Time: Time at MIF. TRAD: training with fixed resistance. EB: training with variable resistance (Elastic band). 5 × 5: five sets of five maximum repetitions.

**Figure 2 sports-09-00142-f002:**
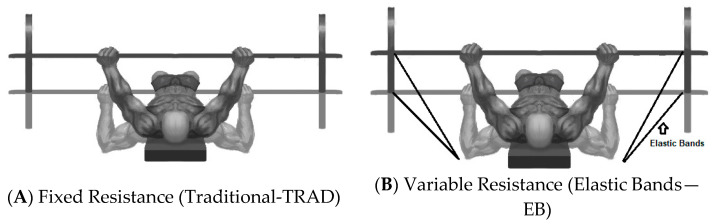
Training Session (**A**) Fixed Resistance (Traditional-TRAD) and (**B**) Variable Resistance (Elastic Bands—EB).

**Figure 3 sports-09-00142-f003:**
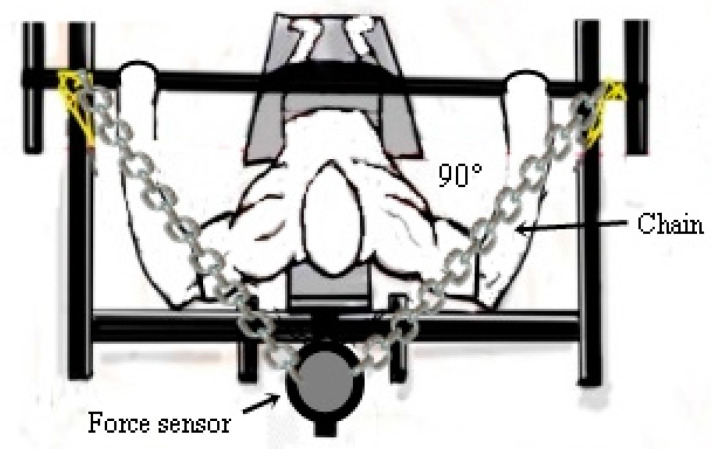
Demonstration of the positioning of the force sensor for evaluation of static force indicators.

**Figure 4 sports-09-00142-f004:**
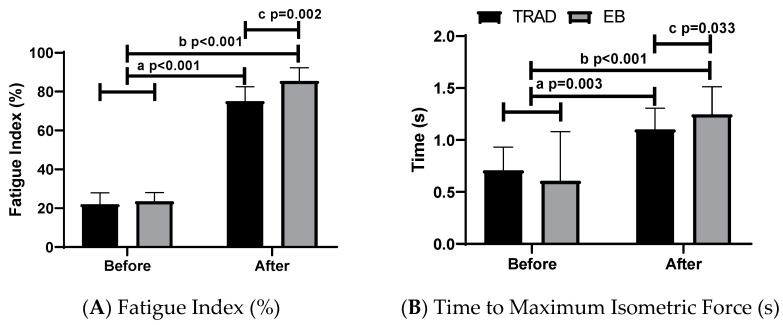
(**A**) Fatigue Index e (**B**) Time to Maximum Isometric Force, in the traditional training (TRAD) and Training with Elastic band (EB). Letters “a”, “b” and “c” indicate *p* < 0.05 (Two-way ANOVA and Bonferroni’s Post Hoc), “^a^” and “^b^” interclass and “^c^” intraclass differences. (**A**) F = 626.97, η2p = 0.983 (very high effect) and (**B**) F = 30.31, η2p = 0.734 (very high effect).

**Table 1 sports-09-00142-t001:** Characterization of the sample.

Variables	Mean ± SD
Age (years)	28.60 ± 7.60
Body mass (kg)	71.80 ± 17.90
Experience (years)	2.57 ± 0.72
1RM test (bench press) (kg)	102.33 ± 21.31
1RM test/body mass ratio	1.43 ± 0.37

**Table 2 sports-09-00142-t002:** 1RM (Kg), RFD (N.s-1), MIF (N), FI (%), and Time (sec) (mean ± SD, 95% CI), in the traditional training (TRAD) and Training with Elastic band (EB).

Variables	1RM (Kg) X ± SD (IC 95%)	MIF (N) X ± SD (IC 95%)	PT (N.m) X ± SD (IC 95%)	RFD (N.s-1) X ± SD (IC 95%)
TRAD Before	98.50 ± 21.37 * (84.92; 112.08)	965.30 ± 209.42 * (832.24; 1098.36)	434.39 ± 94.24 * (374.51; 494.26)	675.28 ± 175.18 *^,^** (563.98; 786.59)
EB Before	98.92 ± 20.80 (85.70; 112.13)	969.38 ± 203.88 (839.84; 1098.92)	436.22 ± 91.75 (377.93; 494.52)	677.21 ± 160.95 (574.94; 779.47)
TRAD After	102.33 ± 21.15 * (88.90; 115.77)	1002.87 ± 207.23 * (871.20; 1134.53)	451.29 ± 93.25 * (392.04; 510.54)	1024.42 ± 305.97 * (830.01; 1218.82)
EB After	101.25 ± 20.09 (88.49; 114.01)	992.25 ± 196.85 (867.18; 117.32)	446.51 ± 88.58 (390.23; 502.80)	895.33 ± 246.61 ** (738.64; 1052.02)
*p* Value	0.018	0.011	0.012	* 0.002, ** 0.016
F	3.337	7.703	7.703	(*) 35.020; (**) 2.920
η2p	0.412 #	0.415 #	0.413 #	0.761 ##

* *p* < 0.05 (Intraclass) ** *p* < 0.05 (Interclass) (Two-way ANOVA, and Bonferroni’s Post Hoc). Effect Size (η2p): # high effect (0.25 to 0.50) and ## very high effect (>0.50). 1RM: 1 Repetition Maximum. MIF: Maximum Isometric Force. PT: Peak Torque, RFD: Rate of Force Development, Time: Time at MIF. EB: Elastic Band. TRAD: Traditional. * Indicates difference in 1RM, MIF, PT, and RFD, before and after traditional training.

## Data Availability

The data that support this study can be obtained from the address: www.ufs.br/Department of Physical Education. Accessed on 26 August 2021.

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
