# Peer review of "Evaluation of Training with Elastic Bands on Strength and Fatigue Indicators in Paralympic Powerlifting"

_sports, 2021, doi:10.3390/sports9100142_

Round 1
Reviewer 1 Report
This manuscript looks at 2 methods of bench press training in Paralympic power lifters. The idea is that the study will better inform coaches about the potential to use variable resistance training with bands. Overall the study seems well designed but the authors could report more clearly about where the subjects came from and inclusion criteria. Referring to the couple of training sessions as an “intervention” is misleading, my understanding is that this is a repeated measures studies with random testing session order.
The writing is not concise which makes understanding harder. The acronyms don’t help understanding much either. Perhaps using letters with some sub-scripts might improve this, for example Tpeak rather than PT.
The introduction, methods, and discussion can be improved. The introduction does not flow so well and does not discuss Paralympic lifting in much detail. Do Paralympic athlete fatigue differently? Would they be expected to? Why would one expect differences? Please expand.
More details are needed in the methods about the participants. Also where were these measurements done? What laboratory? Specific Inclusion/Exclusion criteria?
The discussion could be expanded to be more relevant to the manuscripts data and less discussion about other studies involving similar subjects; I didn’t understand what was similar and I am assuming able-bodied studies? The important indications mentioned need to be detailed more so the reader can understand.
Abstract
line 44: what does “their performance as force indicators” mean?
Line 48: referring to this research as an intervention is confusing.
Line 54: “moments” is a confusing word. Do you mean time moment or torque moment?
Line 63: ordinary(?)
Line 67: washers?
Line 68: angulation? What about using “position” or work word angle in there.
Methods
Line 107 – “Eligibility criterion was to have participated in all intervention session” – this is not the right wording. What were the inclusion criteria and where/how were these Participants recruited? Focus the information on the participants together then add the bit about ethics later – at the moment it is spread out.
Figure 1 is good. A photo of the force measurement set up and elastic bands would also be useful.
Line 160 – was this a scale that weighed them in wheelchairs? If so please simplify wording.
Line 217 – average resistance 20% 1 RM over the entire ROM? Is this correct? What was resistance a peak load then? Regardless of fatigue … would RFD be affected by the kinetics of the setup and load?
Results
I find the layout of Table 2 confusing. Why is there only 1 p comparison per column? Perhaps it is just me, but I would organize so I can see before and after easily for each condition.
Measurements to 2dp (in the table and elsewhere) are not necessary.
Discussion
Line 266 – are you saying 1 session improved strength? I don’t think this is justified.
Line 272 – how did the time of the intervention vary? I cant find that mentioned in the results. So how much difference?
Line 282 – “currents?” like electrically stimulation?
Line 298 – is “interesting” the best word? Applicable?
I am not clear whether fatigue is a good thing or not from your manuscript.
I think there are more limitations to this manuscript than detailed in the paragraph on limitations. The ones mentioned are not discussed in much detail. What IPC class did these athletes come from and how would this make a difference? This should be discussed more detail and would be of interest to readers.
Author Response
Thank you for the considerations presented and the adjustments pointed out are in the manuscript. To differentiate what the reviewers indicated, we placed yellow overlay for reviewer 1, green for reviewer 2 and blue for reviewer 3.

Reviewer 2 Report
Introduction
Please, this part doesn't explain what the fatigue index is. Moreover, there is no mention of speed, only strength. Also, briefly describe what elastic bands do in terms of physiology e.g., induction of high variations of stimuli provokes neural adaptations
Line 97-99 There is no speed in the aim
Methods
I suggest that authors write an "Intervention "paragraph as that is fundamental to this investigation.
Statistics
Please clearly write what is what in ANOVA (2x2) with factors.
Results
Table 2 is a complete mess. It is really hard to follow what represents what. I suggest authors change it and present results with F statistics and Post hoc analysis properly. Also, I assume that normality wasn't violated as that is not stated in the results.
Author Response

(The authors gave the same response as above.)

Reviewer 3 Report
General comments:
As the paper entails with a specific type of variable resistance training (elastic bands), I would change the title by referencing directly to elastic bands rather than to variable resistance. The paper requires an extended English editing.
Specific comments:
- line 65: you should define here, briefly, what variable resistance training is.
- line 67: what is “washers”?
- line 67-68: this sentence is partially incorrect. Just say that the force exerted by a muscle varies according to the joint range of motion as the length of the arm of the muscle force varies according to the joint’s angular position.
- line 69-70: please revise the English form of this sentence.
- line 89-90: you should define here what “fatigue index” is (as you probably do in the methods section), otherwise just refer in general to “fatigue”.
- line 93: “could help coaches IN determinING…”
- line 98: I’ve never heard of “invariable resistance”. The term “invariable” should be replace by “fixed”. Furthermore, it’s not only a matter of variable Vs. fixed resistance but you should specify that the comparison with the fixed resistance has been made with an isometric approach rather than with the classical dynamic way of performing weight resistance training (i.e., ecc-conc repetitions). This is very important to stress in this point of the paper. So you should say “…using elastic bands vs. a training session with fixed resistance in isometric mode”. The reason why you chose isometric contraction is something you should justify in the discussion section. And you should also discuss why you did not use the consecutive repetitions method (dynamic).
- line 107: the eligibility criteria are not very clear. What do you mean with “intervention session”? You should also specify that none of the PP were familiar to variable resistance training.
- line 195: why “dynamic”? It’s just a direct method for determining 1 RM.
- line 201: as several factors related to the execution of the test is crucial for the reliability of the obtained data (you should acknowledge here the following work: https://dx.doi.org/%2010.5812/asjsm.15590) you should stress here that you made sure that the bar height (i.e., elbow extension angle) was kept the same for all the three performed attempts. The same applies when measuring force with the elastic bands.
- line 212-217: this setup is crucial for the reliability of the experiment. Please add some drawings/pictures if possible. Furthermore, you should specify anyway that the experiment with the elastic bands was performed using the same force sensor as used with the isometric test.
- line 270: what do you mean her for “resistance variable”? You mean “variable resistance”?
- Discussion section: as previously said, you should comment on why you used an isometric test modality for your comparison against elastic bands, and what do you expect when using traditional weight resistance training.
Author Response

(The authors gave the same response as above.)

Round 2
Reviewer 1 Report
The article is improved but I have some small corrections:
Line 68 - capitalize start of sentence.
Line 69-70: rewrite this sentence so the improved performance comes earlier. Currently it is understandable but not good grammar.
Line 113: That is not an exclusion criteria. It's worth writing but this is about what data you choose to analyze. I would change and put this sentence in the data analysis section.
Exclusion criteria are defined as features of the potential study participants who meet the inclusion criteria but present with additional characteristics that could interfere with the success of the study or increase their risk for an unfavorable outcome.
Line 383 - OK, I understand now. A limitation was that the participants had the broad range of disabilities which fits into the IPC functional classification.
Author Response
I am very grateful for the considerations and the adjustments suggested were all accepted and placed in the text.

Reviewer 3 Report
dear authors, thanks you for being amenable to my suggestions. Just one thing: at lines 203-204 I would have preferred you stressed the fact that you checked not only the "angle" (which is also a poor expression) but its consistency over the three different attempts (please use "trials" rather than "attempts" as this word sounds like you were not sure that the participant succeeded in performing the task). At lines 203-204 I would have wrote something like "The consistency between the angle of flexion of the elbow at the starting point over the three different trials was checked with.....". Here I also suggested to aknowledge the following paper: https://dx.doi.org/%2010.5812/asjsm.15590 or any other paper that discuss on the requirements that need to be respected when performing such testing.
Author Response

(The authors gave the same response as above.)
